# Four *Mx* Genes Identified in *Andrias davidianus* and Characterization of Their Response to Chinese Giant Salamander Iridovirus Infection

**DOI:** 10.3390/ani12162147

**Published:** 2022-08-21

**Authors:** Yan Meng, Yuding Fan, Nan Jiang, Mingyang Xue, Yiqun Li, Wenzhi Liu, Lingbing Zeng, Yong Zhou

**Affiliations:** Yangtze River Fisheries Research Institute, Chinese Academy of Fishery Sciences, Wuhan 430223, China

**Keywords:** *Andrias davidianus*, Chinese giant salamander, myxovirus resistance, *Mx*, characterization, Chinese giant salamander iridovirus (GSIV) infection

## Abstract

**Simple Summary:**

*Andrias davidianus* is one of the largest amphibian species in the world. To improve our understanding of Myxovirus resistance (*Mx*) genes in amphibians, particularly their function in virus infection, we cloned four full-length *A. davidianus* (*adMx*) cDNA sequences and characterized these genes by bioinformatics analysis and quantitative expression techniques. The four *adMx* genes ranged from 2008 to 2840 bp in length, and their conserved protein domains included the signature architectural feature of the dynamin superfamily. Deduced amino acid sequences exhibited relatively high sequence identity with *Mx* proteins from other vertebrates and phylogenetic analysis revealed close clustering with fish. The four *adMx* genes were broadly expressed in healthy *A. davidianus*, but differentially expressed in the spleen following Chinese giant salamander iridovirus (GSIV) infection. These findings imply that the *adMx* genes share major sequence and protein structures and similar functions with those of other species.

**Abstract:**

Amphibians, including *Andrias davidianus*, are declining worldwide partly due to infectious diseases. The Myxovirus resistance (*Mx*) gene is a typical interferon (IFN)-stimulated gene (ISG) involved in the antiviral immunity. Therefore, knowledge regarding the antiviral immunity of *A**. davidianus* can be used for improved reproduction in captivity and protection in the wild. In this study, we amplified and characterized four different *A. davidianus* *Mx* genes (*adMx*) and generated temporal mRNA expression profiles in healthy and Chinese giant salamander iridovirus (GSIV) infected *A. davidianus* by qualitative real-time PCR (qPCR). The four *adMx* genes ranged in length from 2008 to 2840 bp. The sequences revealed conserved protein domains including the dynamin superfamily signature motif and the tripartite guanosine-5-triphosphate (GTP)-binding motif. Gene and deduced amino acid sequence alignment revealed relatively high sequence identity with the *Mx* genes and proteins of other vertebrates. In phylogenetic analysis, the *adMx* genes clustered together, but also clustered closely with those of fish species. The four *adMx* genes were broadly expressed in healthy *A. davidianus*, but were differentially expressed in the spleen during the GSIV infection. Our results show that the *adMx* genes share major structural features with their homologs, suggesting similar functions to those in other species.

## 1. Introduction

Viral infection stimulates host cells to produce and secrete interferons (IFNs). Interferon (IFN)-mediated antiviral responses are crucial to host defense against viral infection [1]. Interferon triggers the intracellular IFN signaling pathway to induce the expression of related genes, known as IFN-stimulated genes (ISGs), leading to antiviral responses and antiproliferative and immune-regulatory states in the host cells. The induced proteins are responsible for trafficking nucleoproteins into the nucleus to directly antagonize viral replication [2]. *Mx* proteins are typical type I IFN-inducible antiviral proteins belonging to the interferon-induced dynamin guanosine 5′-triphosphatase (GTPase) family. These enzymes consist of an N-terminal GTPase domain, a bundle signaling element and a C-terminal stalk, responsible for oligomerization, viral target recognition, and inhibition of virus replication [3,4]. *Mx* genes have been identified in vertebrates ranging from fish to mammals, with a broad spectrum of antiviral activities not only against RNA viruses, but also DNA viruses belonging to different taxonomic groups [5]. *Mx* proteins can exist both in cytoplasmic and nuclear forms [2]. Most vertebrates express the *Mx1* and *Mx2* analogs of the gene [6], with variation in the numbers in the genomes of different species. Chickens carry one *Mx* gene [7], while pigs carry both *Mx1* and *Mx2,* and humans carry *MxA* and *MxB* [8]. In fish, two *Mx* genes have also been identified in the European sea bass (*Dicentrarchus labrax*) [9], three (*SauMx1*, *SauMx2,* and *SauMx3*) in the gilthead seabream (*Sparus aurata*) [10], seven in the zebrafish (*Danio rerio*) [11], nine in both the Atlantic salmon (*Salmo salar*) [12] and rainbow trout (*Oncorhynchus mykiss*) [13].

The specific antiviral activity of *Mx* proteins in different animals has been reported previously [14,15,16,17,18,19]. *Mx* was first found to inhibit replication of influenza virus (a pathogenic RNA virus) [20], but more recent studies have demonstrated antiviral activity against DNA viruses. For example, *MxA* inhibited replication of the African swine fever virus (ASFV, a large double-stranded DNA) and blocked viral late gene expression by recruitment of *MxA* protein to perinuclear viral assembly sites [21]. As a host restriction factor, *Mx2* affected hepatitis B virus replication in humans [22]. The *SauMx1* and *SauMx2* proteins of Gilthead seabream showed antiviral activity against lymphocystis disease virus infection (LCDV, belonging to the *Iridoviridae* family) [10]. *Andrias*
*davidianus*, which is one of the largest amphibian species in the world, is a significant species in terms of biological evolution and nutritional value. Since the natural populations of amphibians are now in decline, partly due to infectious diseases [23], *A**. davidianus* is now farmed in many locations in China; however, infectious diseases have emerged with the development of artificial breeding. The double-stranded DNA viruses Chinese giant salamander iridovirus (GSIV) or *A. davidianus* ranavirus (ADRV) belonging to the *Ranavirus* genus are important viral pathogens that spread widely, causing significant economic losses in *A. davidianus* farming [24,25,26]. Current strategies for the prevention and treatment of these infections are limited. In addition, our knowledge is limited regarding the genetic evolution of this ancient species, particularly its immune system. Therefore, an in-depth understanding of the antiviral immunity of *A. davidianus* will be useful for improving reproduction in captivity and also for providing protection in the wild. In this study, we identified and characterized four full-length *Mx* cDNA sequences from *A. davidianus*. Moreover, we performed phylogenetic analysis and determined the tissue distribution and expression profiles in the spleen after GSIV infection.

## 2. Materials and Methods

### 2.1. Animals and Sample Collection

Healthy *A. davidianus* (n = 40; length, 18 ± 2 cm; body weight, 100 ± 10 g) were obtained from the experimental farm of Yangtze River Fisheries Research Institute, Chinese Academy of Fishery Sciences. The animals were maintained in tanks at approximately 20 °C and fed daily with diced fish meat for 2 weeks before use in experiments. Individuals were anesthetized using tricaine methane sulfonate (MS-222, Sigma, St. Louis, MO, USA). Heart, liver, lung, skin, kidney, spleen, and muscle tissues were collected and stored in liquid nitrogen. All animal handling and experimental procedures were performed according to requirements of the Institutional Animal Ethics Committee.

### 2.2. RNA Isolation and Gene Cloning

Spleens were collected from healthy *A. davidianus* individuals and total RNA was extracted using TRIzol LS reagent (Ambion, Austin, TX, USA), according to the manufacturer’s instructions. Then, total RNA was used as a template for cDNA synthesis using the PrimeScriptTM RT Reagent Kit with gDNA Eraser (Takara, Dalian, China), according to the manufacturer’s instructions. The open reading frames (ORF) of *Mx* genes were first determined by sequencing of the *A. davidianus* transcriptome [27]. The specific primers for the 3′ untranslated region (UTR) and 5′ UTR (Table 1) were designed based on the ORF sequences of *Mx* genes, and amplified by the rapid amplification of cDNA ends (RACE) using a SMART RACE cDNA Amplification kit (Clontech, Mountan View, CA, USA). The amplification products were detected, purified, and cloned into the pMD18-T vector (TaKaRa, Dalian, China) for Sanger sequencing by Wuhan Tianyi Huiyuan Biotechnology Co., Ltd. (Wuhan, China). The ORF domain and the 3′-RACE and 5′-RACE sequences were aligned by SeqMan (version 7.1.0, DNASTAR software; https://www.dnastar.com/; accessed on 1 May 2022) to generate the full-length cDNA of *Mx* genes. The amino acid sequences encoded by these genes in other species were obtained from the National Centre for Biotechnology Information (NCBI) GenBank database. Multiple *adMx* protein sequences were aligned using the Clustal*W* 7.0.26 and TEXshade 1.26 software. The molecular phylogenetic tree was constructed by Molecular Evolution Genetics Analysis (MEGA 7.0) software using the neighbor-joining method (NJ).

### 2.3. Bioinformatics Analysis

The nucleotide and protein sequence similarities were evaluated through BLAST alignment (http://blast.ncbi.nlm.nih.gov/Blast.cgi; accessed on 01 May 2022). Multiple alignments of *Mx* sequences were performed using Clustal *W* in MEGA 7.0. The molecular weight (MW) and isoelectric point (*pI*) were estimated by EditSeq (DNASTAR, Inc.). The signal peptide was identified by SignalP (http://www.cbs.dtu.dk/services/SignalP; accessed on 01 May 2022) and the protein transmembrane region was identified by TMPred (http://www.ch.embnet.org/software/TMPRED_form.html; accessed on 01 May 2022). The subcellular localization and functional domain of the protein were predicted by Cell-PLoc 2.0 (http://www.csbio.sjtu.edu.cn/bioinf/Cell-PLoc-2/; accessed on 01 May 2022) and PROSITE (https://prosite.expasy.org/cgi-bin/prosite/; accessed on 01 May 2022), respectively. A phylogenetic tree was constructed by MEGA 7.0 using the NJ method with 1000 bootstrap replicates.

### 2.4. Tissue Distribution of the adMx Gene

To investigate the expression profiles of the *adMx* gene in different tissues of *A. davidianus*, RNA was extracted from different tissues of *A. davidianus* (n = 4) and first-strand cDNA was synthesized according to the methods described in Section 2.2 and Section 2.3, respectively. The qPCR primers for amplification of *adMx1*, *adMx2*, *adMx3,* and *adMx4* were designed based on the cloned sequences (Table 1). *EF1-α* was previously verified as the optimal internal control in *A. davidianus* gene expression analysis [28]. Therefore, we designed EF1-αF and EF1-αR primers as an internal control (Table 1). The qPCR reactions were carried out using SYBR^®^ Select Master Mix (2×) (TaKaRa), according to the manufacturer’s instructions with the following reaction system: 10 μL qPCR Mix, 2 μL cDNA, 1 μL each primer, and 6 μL ultrapure water. qPCR was performed on a Rotor-Gene 6000 Real-Time PCR system (Qiagen, Duesseldorf, Germany) as follows: 10 min at 95 °C, 40 cycles of 95 °C for 15 s, 57 °C for 30 s, and 60 °C for 5 min. Relative expression was determined using the 2^−∆∆CT^ method.

### 2.5. Virus Infection and the adMx Expression in Spleen

Thirty-six animals were randomly divided into experimental and control groups. The experimental group was inoculated intraperitoneally with 1 mL GSIV containing 1 × 10^7^ TCID_50_/mL [26]. The control group was injected with the same volume of Dulbecco’s phosphate buffered saline (Sigma). All treated animals were maintained in tanks at 20 °C and fed with fish pieces. Spleen tissue samples were collected at 0 (control), 12, 24, and 48 h after inoculation. Total RNA was extracted from these tissue samples and stored at −80 °C. The qPCR detection of *adMx1*, *adMx2*, *adMx3,* and *adMx4* was performed using primers designed according to their cDNA sequences (Table 1), as described in Section 2.3. At least three biological replicates were included for each test.

### 2.6. Statistical Analysis

All data are expressed as mean ± standard error (SE) using SPSS 22.0 (IBM SPSS, New York, NY, USA). Gene expression was compared by one-way ANOVA followed by the Duncan test using the SPSS software package. A *p*-value of *p* < 0.05 was considered statistically significant.

## 3. Results

### 3.1. Sequence and Domain Architecture Analysis

Four full-length *A. davidianus Mx* gene cDNA sequences (*adMx1, adMx2*, *adMx3,* and *adMx4*) were cloned according to previous transcriptome data using the RACE method. The *adMx1* cDNA was 2808 bp in length encoding a putative protein of 671 amino acids (AA) protein. The *adMx2* cDNA contained 2635 nucleotides encoding a 363-AA protein. The *adMx3* cDNA was the longest sequence composed of 2840 nucleotides encoding a protein of 703 AAs from nucleotides 125 to 2236. The *adMx4* cDNA sequence was the shortest at 2008 bp encoding a 628-AA putative protein. Details of the cDNAs and relative indexes including deduced AA length, MW, and isoelectric point (*pI*) are shown in Table 2. BLAST analysis of the amino acid sequences of the four *adMx* proteins (*adMx1*, *adMx2*, *adMx**3,* and *adMx**4*) showed the highest degree of homology with the *Mx* protein sequences of *Pelodiscus sinensis* (94%, XM_025188401.1), *A. davidianus* (95.8%, KM389533.1), *Pelodiscus sinensis* (94%, XM_014577203), and *Pelodiscus sinensis* (99%, XM_006130518). The cDNA sequences of the *adMx* genes were deposited in GenBank database under the accession numbers *adMx1* ON661517, *adMx2* ON661518, *adMx3* ON661519, and *adMx4* ON661520, respectively.

According to the domain architecture analysis, no signal peptide or protein transmembrane region was located in any of the *adMx* proteins. The putative subcellular localization indicated that, with the exception of *adMx2*, all the *adMx* proteins (*adMx1*, *adMx**3,* and *adMx**4*) were cytoplasmic. The *adMx1*, *adMx**3,* and *adMx**4* protein sequences contained the dynamin-type guanine nucleotide-binding (G) domain signature sequence (LPRGSGIVTR), the tripartite GTP-binding domain (GTPase domain) at the N-terminal end combined with the dynamin family signature, a middle domain (MD), and a C-terminal GTPase effector domain (GED) characterized by a conserved leucine zipper (LZ). With the exception of the MD and GED domains, these domains were also observed in *adMx2*. In addition, the tripartite guanosine-5-triphosphate (ATP/GTP)-binding motifs (GDQSSGKS, DLPG, and TKPD) were all conserved in all four *adMx* proteins. These represent the typical amino acid domains of the dynamic family proteins. Multiple sequence alignment of the *adMx* proteins and their structural characteristics are shown in Figure 1.

### 3.2. Phylogenetic Analysis

A phylogenetic tree was constructed to analyze the relationship of the four *adMx* genes with other *Mx* genes isolated from the mammalian, amphibian, reptile, bird, and fish species. All 50 selected *Mx* protein sequences were divided into different clades, with the sequence of *Mx* Haliotis discus discus (ABI53802.1) as an outgroup (Figure 2). *Mx* from different mammal species formed a cluster, which clustered with one clade including birds (*Gallus gallus*, *Anas platyrhynchos*) and reptiles (*Pelodiscus sinensis*, *Chelonia mydas*) at the top of the phylogenetic tree, which generated group A. All the selected sequences from fish were divided into two groups. One group formed an independent clade of fish (group D) and the other clustered with sequences from *A. davidianus* (*adMx1*, *adMx2*, *adMx**3*, *adMx**4,* and AKA60784.1), forming the group B. Frogs including the *Xenopus tropicalis* and *Xenopus laevis* showed distinct differences compared with *A. davidianus*, and mammalian, bird, reptile, and fish species, to form a separate branch classed as group C in the phylogenetic tree.

### 3.3. Expression of adMx Genes in Normal Tissues

The expression profiles of the four *adMx* genes in different tissues from A. davidianus were analyzed by qPCR (Figure 3). All the *adMx* genes were expressed in the liver, spleen, kidney, skin, muscle, heart, and lung of A. davidianus. The *adMx1* gene was expressed at higher levels in skin and heart, while there were no differences in the expression levels in the liver, spleen, kidney, muscle, and lung (Figure 3A). The highest *adMx2* expression was detected in the spleen, with the lowest levels in skin and liver (Figure 3B). The highest *adMx3* expression was also detected in the spleen, with the lowest in the skin (Figure 3C). There were no significant differences in *adMx4* expression in the tissues, although the levels were slightly increased in the spleen and lung (Figure 3D). Therefore, these results indicate widespread, but variable expression of the *adMx* genes in all the selected tissues.

### 3.4. Expression of adMx Genes in the Spleen Following Treatment with GSIV

The expression of *adMx1*, *adMx2*, *adMx3*, and *adMx4* in the spleen at different time-points after GSIV infection was analyzed by qPCR. Compared with the control, there was no significant difference in the *adMx1* transcript level at 12 h post-infection, while the levels increased significantly at 24 h, followed by a slight decrease at 48 h post-infection (Figure 4A). The *adMx2* transcript levels showed irregular fluctuation over time post-infection (Figure 4B). The *adMx3* expression levels reached peaks at 12 h post-infection, while they showed low expression at 24 and 48 h after GSIV infection (Figure 4C). The transcript levels of *adMx4* increased at 12 and 24 h post-infection, and then decreased at 48 h (Figure 4D).

## 4. Discussion

*Mx* is an interferon-induced GTP-binding protein responsible for a specific antiviral state against a broad spectrum of viral infections in vertebrates [29]. Given the importance of the *Mx* gene in antiviral immunity, we amplified and characterized four full-length cDNA sequences of *Mx* genes (*adMx*) from the *A. davidianus*.

Gene duplication and amino acid substitution are two types of genetic variation that occur in antiviral genes to inhibit emerging pathogens in different species [5]. As an important antiviral ISG, the *Mx* genes have been investigated in many vertebrate genomes and diverse isoforms have been identified. Studies have shown that multiple copies of *Mx* genes are closely linked and may have arisen from local gene duplications in mammals and teleosts [13]. Furthermore, *Mx* genes have been found to exist with variable copy numbers in disparate species [9], including fish, with relatively high copy numbers (from two to nine), whereas the numbers are relatively small in mammals and birds (two copies in mammals and one in birds). In this study, we identified four *adMx* cDNA sequences in *A. davidianus*. The copy numbers of these genes in *A. davidianus* were higher than those in mammals and birds, but lower than those in some fish, indicating that the lower vertebrates have more *Mx* genes than higher vertebrates. In addition to the innate immune system, higher vertebrates have developed an efficient adaptive immune system during evolution. Since the *Mx* protein is a component of the innate immune system, higher vertebrates have fewer *Mx* genes in their genomes. These species-related differences in the numbers and functions of *Mx* proteins as antiviral effectors reflect the long-term evolution of the host immune system and viral immune evasion [19]. Compared with the higher vertebrates, lower vertebrates appear to possess more innate immune system-related molecules or analogues to defend against microbial invasion, although research in *A. davidianus* is relatively limited. Liu et al. [30] cloned a 2848-bp *A. davidianus Mx* gene cDNA sequence encoding 703 AAs, while Chen et al. [31] identified a 2562-bp *Mx* gene cDNA sequence (KM389533.1) containing 363 AAs from the same species. Although *adMx**3* and *adMx2* were predicted to contain the same number of AAs in our study, sequence comparisons revealed marked differences in the AA content of the proteins. In contrast, there was only one base difference between the *adMx**3* and 363 AAs sequence (AKA60784.1) reported by Chen et al. Whether these are individual or species variations in *adMx* genes remain to be clarified.

In this study, a total of 50 *Mx* protein sequences were divided into four different clades, with the exception of the *Mx* protein of *Haliotis discus discus*, which formed an outgroup. The four *adMx* proteins identified in this study first clustered together in the evolutionary tree, and then combined with the genes in fish (*Danio rerio MxE*, *Danio rerio MxC*, *Cirrhinus mrigala Mx*, *Oncorhynchus mykiss Mx2*, *Oncorhynchus mykiss Mx,* and *Carassius auratus Mx3*). The *Mx* proteins of birds (*Gallus gallus*, *Anas platyrhynchos*) and the reptiles (*Pelodiscus sinensis*, *Chelonia mydas*) belonged to one clade and then combined with those of mammalian species. Finally, the four different taxonomic statuses formed a cluster. Since *A. davidianus* undergoes a transition from aquatic life to terrestrial life, this result seems to be consistent with its taxonomic classification status in the transition stage between aquatic and terrestrial vertebrates. Strangely, the *Xenopus tropicalis*, *Xenopus laevis,* and *A. davidianus* did not cluster together, despite belonging to the amphibian class. The four sequences from *Xenopus tropicalis* and *Xenopus laevis* clustered to form a branch that differed from the other clades of fish, reptiles, birds, and mammals.

*Mx* is an interferon (IFN)-inducible dynamin-like GTPase protein. Similar to most dynamin-like GTPases, *Mx* proteins are composed of an N-terminal GTPase (G) domain, a middle domain (MD), which is known as a central interactive domain (CID), and a C-terminal GTPase effector domain (GED). The GTPase domain, including the GTP-binding motif and dynamin signature, is the most conserved structural feature of *Mx* proteins and mediates the dynamin-like GTPase properties of *Mx* proteins. This GTPase activity is required for their antiviral activity. The MD is rich in α-helices and important for oligomerization and viral target recognition. The GED can fold back to join the N-terminal GTP-binding domain via the conserved C-terminal leucine zipper to establish the enzymatically active center of *Mx* protein, and enhance the GTPase activity [32,33,34]. In this study, we demonstrated that the *adMx1*, *adMx**3,* and *adMx**4* contained a GTPase domain, as well as a MD and GED domain, while *adMx2* possessed only the GTPase domain. The *adMx2* gene is similar to the other three genes in terms of nucleotide sequence, although the ORF domain is shorter. Previous studies indicated the existence of more than ten exons in the *Mx* mRNA sequences of many species [19]; therefore, we speculated that this variation is due to alternative splicing. The various structural, biophysical, and biochemical properties of the dynamin superfamily reflects their distinct cellular functions [35], while the subcellular localization of a protein can also determine its function. According to the subcellular localization analysis, *adMx2* was located in the cell nucleus; however, *adMx1*, *adMx**3*, and *adMx**4* were predicted to be located in the cytoplasm. The *Mx* proteins accumulate rapidly in the nucleus or cytoplasm, self-assemble in oligomers, and interfere with viral replication during viral infection of host cells [36]. *Mx* proteins in the nucleus or cytoplasm possess distinguishing antiviral functions, at least in part of their antiviral spectrum [16], implying that *adMx2* has different antiviral functions compared with the other three *adMx* proteins.

We investigated the distribution of *adMx* expression in normal tissue and changes in response to GSIV infection. We found that the *adMx* genes were broadly expressed in the seven tissues analyzed in normal *A. davidianus*, although their expression profiles varied in different tissues. With the exception of *adMx1*, the other three *adMx* genes were expressed at high levels in the spleen. As the largest secondary lymphoid organ, the spleen is the site of a wide range of immunologic processes, including those related to both innate and adaptive immune responses, alongside its roles in hematopoiesis and red blood cell clearance [37,38]. In the current study, we showed that *adMx* transcript expression changed over time after GSIV infection of *A. davidianus*. Compared with the control, only *adMx4* was significantly upregulated at 24 h post-infection followed by decreased expression to 48 h post-infection. The expression of *adMx2* fluctuated post-infection, with increased levels detected at 12 h, followed by a decrease in the level at 24 h and peak expression at 48 h post-infection. In a previous study of the response to GSIV infection, the expression level of one *A. davidianus Mx* gene was found to increase at 6 h post-infection in the kidney, spleen, and muscle and peaked at 48 h, while in the levels in muscle, the cell lines were not upregulated until 72 h post-infection [30]. It was also reported that another *Mx* gene cloned from *A. davidianus* was upregulated at 48 h in gsIFN-overexpressing cells [31]. Although comparisons revealed differences in the *Mx* expression profiles, the general patterns were similar. In fact, the *Mx* gene isoforms showed different expression profiles in the same species following virus infection. Following infection of gilthead seabream with nervous necrosis virus (VNNV), three *Mx* genes displayed differences in expression in terms of tissue distribution, time course, and level of induction. In brain, *Mx2* showed the strongest and quickest response, with significant induction as early as 24 h post-infection, whereas *Mx1* and *Mx3* could not be detected before day 5. In head kidney, *Mx2* also showed the strongest and quickest response, with maximum expression at 24 h post-infection, followed by *Mx1*, which showed a weak response detected at 24 h post-infection. In contrast, *Mx3* was undetectable until day 30 post-infection. Furthermore, the expression of the *Mx* genes fluctuated during the antiviral response [10]. *Mx* gene expression was shown to increase significantly after type I IFN expression was induced in *Xenopus laevis*, and substantially reduced frog virus 3 (FV3) replication both in vitro and in vivo [39]. Overexpression of the mullet fish *Mx* gene (*MuMx*) confirmed the significant inhibition of viral hemorrhagic septicemia virus (VHSV) transcripts, while eliciting crucial antiviral functions against viral antigens [29]. The *SauMx1* and *SauMx2* of gilthead seabream have antiviral activity against lymphocystis disease virus (LCDV), although *SauMx3* did not inhibit LCDV replication in CHSE-214 cells [10]. Therefore, these studies of the response of *Mx* gene expression to the virus indicate that the functions of *Mx* vary with species and isoforms.

## 5. Conclusions

In this study, we have expanded our understanding of the structure, expression, and function of *Mx* genes and proteins in antiviral immunity in amphibians. We identified four *Mx* genes in *A. davidianus* and characterized their response to GSIV infection. These results enrich our knowledge of the antiviral function of *Mx* genes in response to DNA viruses.

## Figures and Tables

**Figure 1 animals-12-02147-f001:**
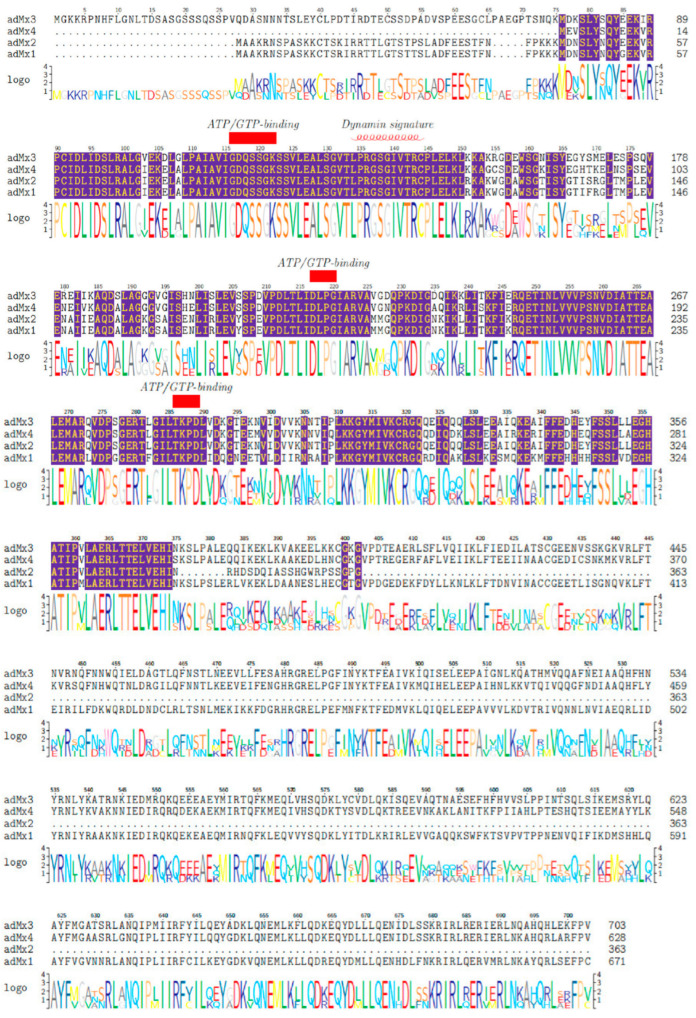
Multiple sequence alignment of the *adMx* proteins. The ATP/GTP binding sequence is indicated with a red frame. The dynamin signature sequence is indicated by a red helix. Conserved amino acids are shaded as indicated with a purple frame. Dots indicate positions of amino acid deletion.

**Figure 2 animals-12-02147-f002:**
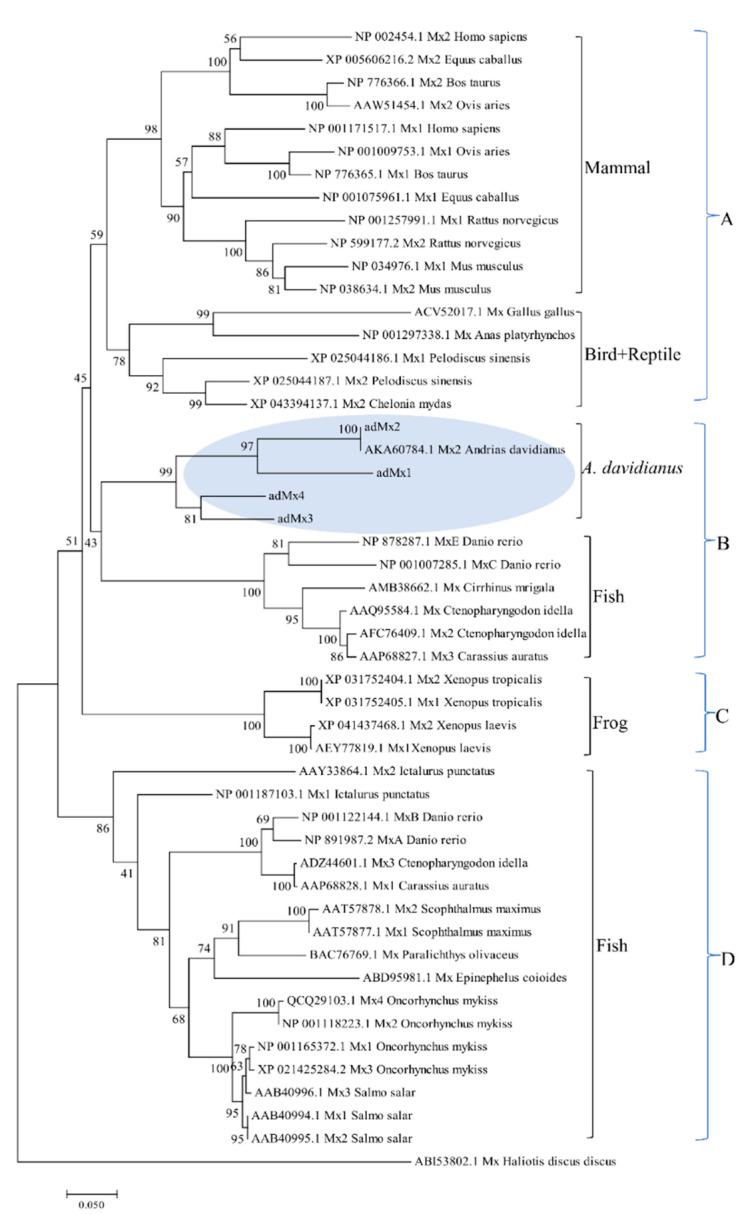
The NJ phylogenic tree of different species based on *Mx* protein sequences constructed using MEGA. Sequence accession numbers used in the analysis are shown. Bootstrap percentages are shown on interior branches with the bootstrap values based on 1000 replications. Bars indicate genetic distances according to the 0.050 scale. (A, B, C, D) represent the four groups. The *adMx1*, *adMx2*, *adMx3*, and *adMx4* proteins are indicated by a pale blue oval.

**Figure 3 animals-12-02147-f003:**
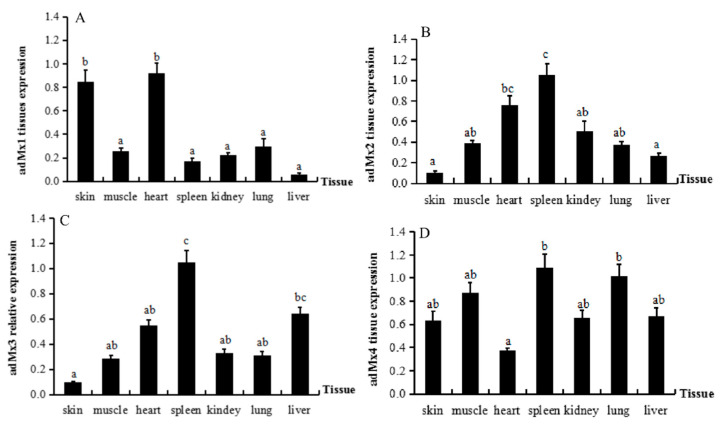
The expression profiles of four *adMx* genes in different tissues from *A**. davidianus*. Data represent the mean ± SE (n = 3). The letters above the columns indicate significant differences (*p* < 0.05). (**A**–**D**) represent the gene expression of *adMx1*, a*dMx2*, *adMx3,* and *adMx4* in different tissues.

**Figure 4 animals-12-02147-f004:**
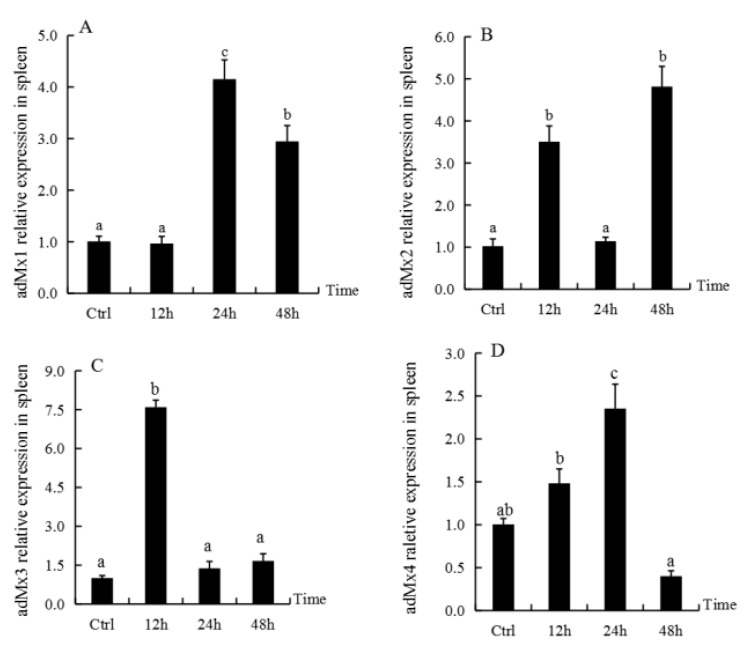
The *adMx1, adMx2, adMx3,* and *adMx4* transcript levels in *A**. davidianus* spleen following GSIV infection. Data represent the mean ± SE (n = 3). Different letters indicate significant differences (*p* < 0.05). (**A**–**D**) represent the gene expression of *adMx1*, *adMx2*, *adMx3,* and *a**dMx4* in spleen treatment with GSIV.

**Table 1 animals-12-02147-t001:** Primers used in this study.

Primer	Sequence (5′–3′)	Purpose
*Mx1* 3P	TGGAGGTTGTTGGAGCACAACAGAA	3′-RACE
*Mx2* 3P	TTGTGGTGCCAAGTAATGTGGATAT	
*Mx3* 3P	GGTTGCTCAAACAAATGCAGAGTCTGAA	
*Mx4* 3P	GCAGCCTGTGGAGAAGACATTTGTTCTAA	
*Mx1* 5P	TCTCTCCGCCGGGATCCACCAGTCG	5′-RACE
*Mx2* 5P	GCAATCCCAGGAAGGTCAATCAGTG	
*Mx3* 5P	CCACATTTCTTTAGTTCCTCTTTCGCTAC	
*Mx4* 5P	ACAAATGTCTTCTCCACAGGCTGCGTT	
*Mx1* qF	ATCCCGCTGAAGAAGGGTTAC	qPCR
*Mx1* qR	CGTTTGCTGCGTCAAGTTTCT	
*Mx2* qF	TTCCCAGAGGCAGTGGTATTG	
*Mx2* qR	ACTTCTAAGGGCATTGTCAGG	
*Mx3* qF	CGAGGAGATGAATGGAGTGGG	
*Mx3* qR	CCACTCCTCCACCAGCCAGA	
*Mx4* qF	AGAGGCACTAGAAATGGCACG	
*Mx4* qR	TGTCCTGGATGTCCTGTTGC	
EF1-αF	GGACAGACCCGTGAACATGC	Internal reference
EF1-αR	CTTCCTTAGTGATCTCCTCGTAGC	

**Table 2 animals-12-02147-t002:** Details of the four *adMx* cDNA sequences and relative indexes.

Gene	Length (bp)	5′-UTR (bp)	ORF (bp)	3′-UTR (bp)	Amino Acids (AA)	MW (Daltons)	pI
*adMx1*	2808	1–54	55–2070	2071–2808	671	76,688.82	8.09
*adMx2*	2635	1–53	54–1145	1146–2635	363	39,786.77	6.53
*adMx3*	2840	1–124	125–2236	2237–2840	703	79,105.06	5.23
*adMx4*	2008	1–38	39–1925	1926–2008	628	71,301.06	6.42

MW: Molecular weight; *pI*: Isoelectric point; ORF: Open reading frame; UTR: Untranslated region.

## Data Availability

The data supporting the findings of this study are available within the article.

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
