# Peer review of "Four Mx Genes Identified in Andrias davidianus and Characterization of Their Response to Chinese Giant Salamander Iridovirus Infection"

_animals, 2022, doi:10.3390/ani12162147_

Round 1
Reviewer 1 Report
The authors have responded to all of the comments and suggestions and the manuscript has been improved significantly. I recommend presenting the manuscript for publication in the current version.
Author Response
We are very grateful to you for the recognition to our paper.
Reviewer 2 Report
Dear authors, I appreciated this work but still the manuscript is far from be ready to be accepted on a journal like Animals. Best of luck
Line 8: Remove “(A. davidianus)”
Line 16: Use abbreviation A. davidianus
Line 20: The abstract should be able to stand alone so you cannot use the abbreviation of the scientific name the first time it is mentioned. In addition, why there is the present tense “undergo”? Cannot understand. Please correct
Lines 20-22: “The A. davidianus undergo 20 transition from the aquatic life to terrestrial life in animals. It has great importance in the vertebrate 21 evolution.”. This is not the background that is necessary in the abstract. State that amphibians are declining worldwide (see amphibian crisis) and mention that also this species is declining too. Thus, knowledge about antiviral immunity of this species can be used to reproduce these animals in captivity and to protect them also in the wild. Re-phrase and modify.
Line 29: “high homology” means nothing since homology is homology. You have homology but it is not high or low. Please change with “sequence identity”.
Line 31: “…with those of fish species”. THOSE missing. Please add, thanks
Line 36: Remove “expression profile” from the keywords. Thanks
Line 54: Mx must be in italics
Line 62: “in species”. What does it mean? This does not sound so scientific since it is obvious it is “in species”. So specify better. Please, also correct the grammar in “firstly found…”
Line 66: missing citation
Line 72: You cannot start a sentence with abbreviated form of a scientific name. Please modify thanks
Line 73: “…and it has been listed as Critically Endangered by the IUCN Red List (Proper citation).”. See https://www.iucnredlist.org/species/1272/3375181. You must add this. Thanks
Line 75: Which scientific value this sentence (In addition, it was considered to have high nutritional value since ancient 75 times.) add to the manuscript?
Line 77: it farmed. Also remove “the” before infectious. Check grammar. Thanks
Line 80: Why are you using past tense? Those viruses are not extinct. Please correct. Same line 86
Line 97: Missing space
Lines-75 72 should be removed. I will ask it again: The species has to be introduced as a species listed as Critically Endangered by the IUCN Red List (PROPER CITATION OF IUCN RED LIST HERE). In addition, before of this background on the “Amphibian crisis” and the need to study their pathogens to protect wild (and eventually captive) populations of these animals must be clearly claimed. “In addition, it was considered to have high nutritional value since ancient times.” Has no sense if it is not claimed that this is pure speculation. There is NO scientific evidence of its value from the nutritional point of view. The scientific name cannot be abbreviated when it is at the beginning of a sentence. Modify accordingly
Line 94: Remove “The”
Line 95: With a mean value a sd/se must always be provided.
Line 96: Missing space
Line 103: Remove “The”
Line 134: genes names must be in italics
Line 151: what does “normally” mean in that sentence?
Line 152: At control? what does it mean? Line 156: What about the technical replicates of the qPCRs?
Line 169: What does mean that there was an initiation codon at nucleotides 125 to 2236?
Line 192, 195, 216-217, 356-358: English
Line 205: Put abbreviation
Line 211: You have to mention here in the caption what the scalebar represents.
Line 216-217: Statistics is missing. Regarding figure 3 how the samples were compared? In addition, graphs have different values on the y-axes and this can lead to confusion.
Lines 242-246, 250, 256, 264: English
Line 247: Genetic innovations? What does it mean?
Line 252: Missing citation
Line 354: during antiviral infection OR in antiviral immunity. The latter is better
Author Response
We are very grateful to you for the recognition to our article and providing these very valuable comments. According to the comments, we have revised them item by item in the revised manuscript.

Reviewer 3 Report
This manuscript by Meng et al identifies four Mx genes in the Chinese giant salamander. Furthermore, experimental infection with Chinese giant salamander iridovirus demonstrated differential expression of the four Mx genes over time in the spleen. This is a well-designed study presenting valuable data for the evolutionary biology and immunology fields; however, the manuscript may be improved with consideration of the following comments.
1. The authors briefly mention the possibility of alternative splicing in the Discussion (line 308). Please expand on the discussion of the 4 identified Mx cDNAs belonging to four separate genes versus alternative splicing products of one or more genes.
2. Figure 2 – Please highlight the Andrias Mx genes on the figure for easier identification.
3. Figures 3 & 4 – Please present the data in these figures as individual data points so that readers can more easily evaluate sample size. Please also provide a more in-depth description of what each of the letters above the columns indicate as significance (comparison between what data points, etc). Please also make the range of the y-axes the same for each panel in each figure (i.e. make all Fig 3 graphs go to 1.4 and all Fig 4 graphs go to 9.0) so that the readers may more easily compare relative gene expression across Mx genes.
Author Response
We are very grateful to you for the recognition to our paper and providing these very valuable comments. According to the comments, we have revised them in the manuscript.

Reviewer 4 Report
Summary:
The manuscript titled “Four Mx genes identified in Andrias davidianus and characterization of their response to Chinese giant salamander iridovirus infection” investigates Mx genes in Andrias davidianus. These genes are important in antiviral activity in its host. This manuscript is generally well written, however, sentence structure and sentence transition can be improved upon to improve the flow of the writing. Methods are well detailed. I have the following comments:
ABSTRACT
Line 23: Consider moving the sentence starting with “It has great importance…” before the sentence “The A. davidianus undergo…” to improve the paragraph flow.
MATERIALS AND METHODS
Line 100: Do you mean the animals were euthanized? Or were the tissues from the animals collected while the animals were still alive? Please clarify how the tissues were humanely collected.
Line 150: Why was GSIV used to study viral infection?
Figure. 3
This image is a little blurry compared to the other figures.
DISCUSSION
Line 334: Please correct repeated word “After After comparison,…”
Line 341: What does “in head kidney” mean?
CONCLUSION
Please elaborate on the benefits of doing additional studies on the adMxs in Ranavirus. How will this further the field of research?
Author Response
We are very grateful to you for the recognition to our paper and providing these very valuable comments. According to the comments, we have revised them in the manuscript.

This manuscript is a resubmission of an earlier submission. The following is a list of the peer review reports and author responses from that submission.
Round 1
Reviewer 1 Report
Generally:
Gene names should be italics – this occurs throughout the manuscript and should be corrected.
There is no information about sex, a number of animals, and relations between them. This should be clearly described in the “Animals and Samples Collection” section. In “Virus Infection and AdMx Expression in Spleen” section – how many individuals were in the control and experimental group? Additionally, this information is necessary for 3.3 and 3.4 sections.
The information about the sequencing method in “RNA Isolation and Gene Cloning” should be completed.
The number of Ethics Committee agreement should be presented.
Punctuation should be checked throughout the manuscript. Commas are missing in many places.
Articles should also be completed. Some examples of the missing or incorrect articles in the text: the spleen (lines 17, 32, 78), an antiviral (line 41), a typical (line 44), a different (line 52), in the cytoplasm (line 153), the Neighbor-joining (line 168), the lowest (line 192) and many more.
Another common mistake is using singular nouns in places where plural nouns should occur, the examples: functions (line 18), states (line 42), viruses (line 50), genes (line 53), indexes (line 142), mammals, birds, reptiles and fishes (line 178), mammals (lines 221, 224), birds (line 225), vertebrates (line 234).
The other inaccuracies and errors in line by line review:
Lines 18 and 34 – “with other species” should be replaced with “in the other species”
Line 19 – sentence “Myxovirus resistance (Mx) gene is a typically interferon (IFN)-stimulated genes (ISGs)…” needs to be rewritten correctly (“typically” should be replaced with “typical”, singular and plural nouns)
Lines 21 and 69 – “evolutionary” should be replaced with “evolution”
Line 22 – “Here” is unnecessary
Line 25 – “under healthy condition” should be changed
Line 29 – “it” should be replaced with “they”
Line 31 – “and then combined with fish” is not clear enough
Line 45 – “protein” should be in the plural form
Line 48 – “of” before “virus” is needed
Line 49 – “with” is unnecessary
Line 56 – when the other species are listed in English please write also D. rerio in English
Line 59 – “inhibiting” should be replaced with “to inhibit”
Line 60 – “It” should be replaced with “Its”
Line 64 – “recruit to form virus factories” do you mean factors?
Line 68 – “which underwent transition” should be replaced with “and underwent a transition”
Line 69 – “in animal evolution” is unnecessary
Line 70 – “as” is unnecessary
Line 71 – “breed” should be replaced with “breeding”
Lines 71-72 – “during the process” is unnecessary
Lines 72-73 – “belonged” should be replaced with “belonging”
Line 86 – “methane sulphonate” the correct spelling is “methanesulphonate”
Lines 91 and 104 – Please write what does it means “normal”
Lines 118-119 – “the control group were injected the same volume” should be replaced with “the control group was injected with the same volume”
Lines 120-121 – please change “0h” for a control
Line 121 – “and” after “collected” is missing
Line 121 – “in -80” should be replaced with “at -80”
Line 128 – the correct name of the test is Duncan, not Duncant
Line 129 – “at” should be replaced with “in”
Line 135 – use “with” after “combined”
Lines 141 and 161 – “detail” should be replaced with “detailed”
Line 142 and Table 2 – the wrong spelling of the word “isoelectric”
Line 143 – “pl” should be replaced with “pI”, large “i”, not “L”
Line 143-144 – “The sequence identify of four adMx protein amino acid” should be replaced with “The sequences identified of four adMx protein amino acids”
Lines 152, 246, and 287– “except” occurs with “for”, not “of”
Line 173 – “in top” should be replaced with “at the top”
Line 175 – from the context, does “dependent” should be replaced with “independent”?
Lines 187 and 250 – “was” should be replaced with “were”
Lines 187-188 – I think here should be “are” after “genes”
Line 189 – “and then” should be replaced with e.g. “however”
Line 189 – after “difference” should be “in”
Line 193 – “about” should be changed with e.g. “in case of”
Line 193 – here is a verb missing after “in spite of”
Line 201 – “to” should be changed with “of”
Line 201 – I think that “After GSIV challenged” is not necessary in this sentence
Line 202 – “or” should be replaced with “and”
Line 202 – please, explain what „dpi” means
Line 204 – the word “challenged” is unnecessary
Line 205 – the plural form will be better than “its”
Line 205 – here is a missing verb after “low expression”
Lines 210, 263, and 309 – Do not start this sentence with “And”
Line 215 – replace “here” with e.g. “in the research”
Lines 221-222 – “As described above…” – the sentence is not clear
Lines 223 and 224 – Do not use the word „members” when you write about the copies
Lines 225-226 – Rewrite the sentence “It was more than mammal…” correctly
Lines 227-228 – the sentence “For high vertebrates…” includes a lot of mistakes: There are no „high vertebrates” but higher vertebrates; except for, not of; inmate means the prisoner, „they” is unnecessary
Line 230 – the genes are countable so we should not use “less” but “fewer”
Line 233 – “defense” should be replaced with “defend”
Lines 235-237 – “The diversity, accuracy and…” – the sentence is not clear and has a few mistakes
Line 246 – the correct spelling is “outgroup”
Lines 251-254 – sentence “Considering the evolutionary relationship…” should be rewritten more clearly
Line 255 – “they” is unnecessary
Line 256 – “and” should be used before “they”
Line 261-262 – “part in” should be replaced with “part of”
Line 268 – “all” seems to be unnecessary
Line 270 – “similar” occurs with “to”, not “with”
Line 271 – “studied” is a wrong form
Line 272 – unfinished sentence
Line 272 – the missing “if” after “wondered”
Line 274 – missing verb after “adMx2”
Line 278 – missing verb after “cells”
Line 278 – wrong spelling of word proteins
Line 281 – missing “of” after “in part”
Line 285 – “demonstrated all of them broadly…” should be replaced with “demonstrated that all of them were broadly…”
Line 288-291 – reduce one „as such” in this sentence
Line 293 – missing verb before “stimulated”
Line 294-295 – please, describe how the expression fluctuated
Line 300 – “but” in unnecessary
Line 302 – not sure if „treatment” is a good word, maybe “infection” will be better?
Line 306 – shouldn’t be here “after” or “in”, not “before”?
Line 306 – does “head kidney” means “the head and kidney”?
Lines 306-307 – “Mx2 was strongest” should be replaced with “Mx2 had the strongest”
Line 307 – “and a maximum” should be replaced with “with a maximum”
Line 308 – is there the end of the sentence after “24 h”?
Line 310 – missing “in” before “Xenopus laevis”
Line 313 – “elicts” should be replaced with “eliciting”
Line 320 – “to viral” should be replaced with “in viral”
Lines 320-323 – from “The results…” till the end of the conclusions the sentences are not clear enough, please rewrite them
Figure 3: Please arrange the tissues in each plot in the same order and pay attention to the same font size in all the plots (also in Figure 4)
References:
In general, please standardize the style of references - when there are 2 author's names, sometimes there is a space between them, and sometimes not
Line 348 – extra space
Lines 352, 354, and 379 – no dots at the end of the sentences
Lines 354, 396, and 399 – no bold font
Line 361 – point 15. written two times
Line 376 – a lack of space
Line 395 – a lack of italics
Reviewer 2 Report
Dear authors, here attached you can find a file with the requested changes. The work is interesting and methods appropriate but the ms need further work to be published.
Best,
the reviewer
